# Anomalous Landau quantization in intrinsic magnetic topological insulators

Su Kong Chong [1,6] ✉, Chao Lei[2,6], Seng Huat Lee [3,4], Jan Jaroszynski [5], Zhiqiang Mao [3,4], Allan H. MacDonald [2] & Kang L. Wang [1] ✉

The intrinsic magnetic topological insulator, $Mn(Bi_{1-x}Sb_x)_2Te_4$, has been identified as a Weyl semimetal with a single pair of Weyl nodes in its spin-aligned strong-field configuration. A direct consequence of the Weyl state is the layer dependent Chern number, $C$. Previous reports in $MnBi_2Te_4$ thin films have shown higher $C$ states either by increasing the film thickness or controlling the chemical potential. A clear picture of the higher Chern states is still lacking as data interpretation is further complicated by the emergence of surface-band Landau levels under magnetic fields. Here, we report a tunable layer-dependent $C = 1$ state with Sb substitution by performing a detailed analysis of the quantization states in $Mn(Bi_{1-x}Sb_x)_2Te_4$ dual-gated devices—consistent with calculations of the bulk Weyl point separation in the doped thin films. The observed Hall quantization plateaus for our thicker $Mn(Bi_{1-x}Sb_x)_2Te_4$ films under strong magnetic fields can be interpreted by a theory of surface and bulk spin-polarised Landau level spectra in thin film magnetic topological insulators.

Magnetic topological insulators (MTIs) and Weyl semimetals have both received a great deal of attention in recent condensed matter physics research[1-3]. Particularly, the intrinsic MTI $MnBi_2Te_4$ has provided researchers with an ideal candidate to study the relationship between topological quantum states and magnetic phases[4-15]. Theoretical predictions[16-19] and recent experimental results[20] show that when its Mn local moment spins are aligned by an external magnetic field, the energy bands of bulk $MnBi_2Te_4$ have a single isolated pair of Weyl crossing points that are close to the Fermi level and therefore can be accessed by controlling the carrier-density.

In this work, we report on thickness-dependent magneto-transport studies of the mechanically-exfoliated $Mn(Bi_{1-x}Sb_x)_2Te_4$ thin flakes with three different Sb concentrations. The Sb substitutions in the $MnBi_2Te_4$ parent compound (i) move the Fermi level of the bulk bands closer to the charge neutrality point (CNP)[20-22], and (ii) modulate the Weyl point separation in momentum space as illustrated in Fig. 1a. We focus here on the Chern insulator states of the spin-moment aligned $Mn(Bi_{1-x}Sb_x)_2Te_4$. We show that Sb substitution extends the surface gap regime to a wider thickness range by suppressing conduction from the trivial bulk bands. Thin films $Mn(Bi_{1-x}Sb_x)_2Te_4$ provide a rich plethora of topologically distinct quasi-two-dimensional (2D) states that includes Chern insulator[4-6,8-10] and axion insulator[5,7,16,23] states. The application of external magnetic fields to $Mn(Bi_{1-x}Sb_x)_2Te_4$, therefore, generates an interplay between Chern gaps and Landau levels (LLs) quantization, allowing us to study rich quantum Hall physics that has not yet been fully explored.

## Results

### $Mn(Bi_{1-x}Sb_x)_2Te_4$ thin film Weyl semimetals

According to density-functional-theory (DFT), the spin-aligned magnetic configuration of $Mn(Bi_{1-x}Sb_x)_2Te_4$ is a simple type-I (or II depending on lattice parameters[16]) Weyl semimetal with Weyl points at $\pm k_w$ along the $\Gamma - Z$ line. As shown in Fig. 1a, the distance between Weyl points ($2k_w$) decreases with Sb fraction $x$. The bulk Hall

[1]Department of Electrical and Computer Engineering, University of California, Los Angeles, CA 90095, USA. [2]Department of Physics, The University of Texas at Austin, Austin, TX 78712, USA. [3]2D Crystal Consortium, Materials Research Institute, The Pennsylvania State University, University Park, PA 16802, USA. [4]Department of Physics, The Pennsylvania State University, University Park, PA 16802, USA. [5]National High Magnetic Field Laboratory, Florida State University, Tallahassee, FL, USA. [6]These authors contributed equally: Su Kong Chong, Chao Lei. ✉e-mail: sukongc@g.ucla.edu; wang@seas.ucla.edu

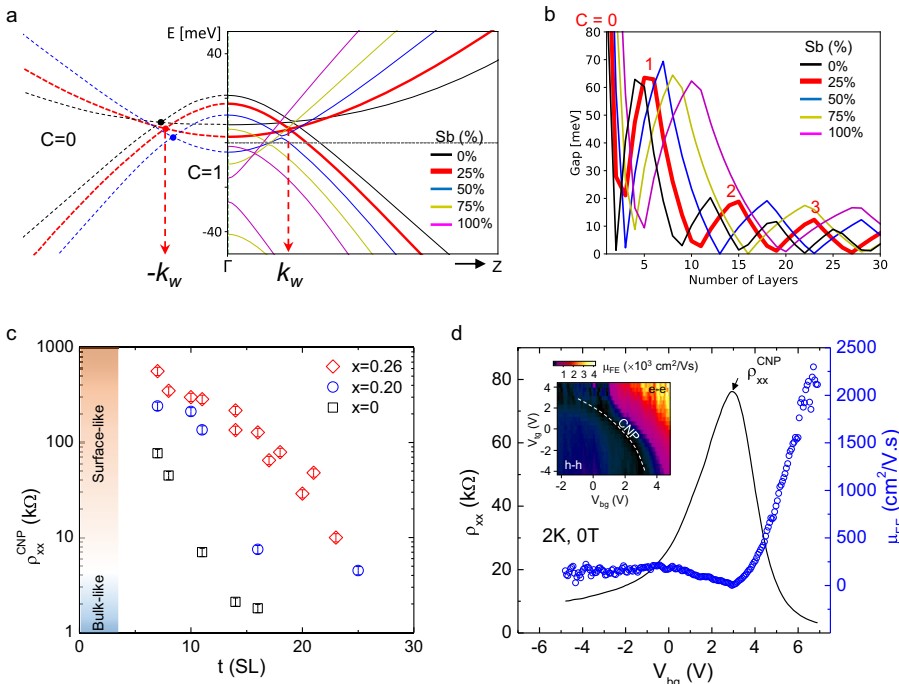

**Fig. 1 | Tunable Chern insulator states and transport properties. a** DFT calculated band structures for spin-aligned $Mn(Bi_{1-x}Sb_x)_2Te_4$ intrinsic magnetic topological insulators at different Sb substitution levels. The positions of the Weyl point $(k_w)$ for $x = 0\%$, $25\%$, and $50\%$ are shown as black, red, and blue nodes, respectively, in the figure. The $C = 0$ and 1 denote the Chern number outside and within the $\pm k_w$, respectively. The solid and dashed lines represent band structure along the $\Gamma - Z$ and $\Gamma - \bar{z}$ lines, respectively. **b** Calculated magnetic exchange gap size as a function of film thickness for the $Mn(Bi_{1-x}Sb_x)_2Te_4$ at different Sb substitution levels. The thickness ranges for $C = 0, 1, 2, 3$ at $x = 25\%$ are labeled in the figure. **c** Resistivity at charge neutrality point $(\rho_{xx}^{CNP})$ plot as a function of flake thickness for

$Mn(Bi_{1-x}Sb_x)_2Te_4$ at Sb concentrations of $x = 0$, 0.20 and 0.26. $\rho_{xx}^{CNP}$ is defined as the gate-dependent resistivity peak measured at a temperature, $T = 2$ K, at zero magnetic field. The error bars in (**c**) are estimated from the uncertainty in the determination of the device geometry. **d** $\rho_{xx}$ plot as a function of backgate voltage for a 18-SL $Mn(Bi_{0.74}Sb_{0.26})_2Te_4$ device measured at temperature, $T = 2$ K, at zero magnetic field. The extracted field effect mobility $(\mu_{FE})$ as a function of backgate voltage is plotted in the secondary $y$-axis on the right. The inset in (**d**) is the $\mu_{FE}$ mapping as a function of dual-gate voltages taken at 2 K. The hole–hole (h–h) and electron–electron (e–e) labels in the inset denote the carriers' conduction of the top-bottom surfaces as controlled by top-bottom gate voltages.

conductivity normalized per layer can be expressed as $\frac{\sigma_{xy}^{3D}}{d} = \frac{e^2}{h} \frac{k_w d}{\pi}$, where $k_w$ is the position of Weyl point and $\pi/d$ is the size of Brillouin zone along the $\Gamma - Z$ line with $d$ as the septuple layer (1SL ≈ 1.4 nm). The corresponding thin films can be viewed as quasi-2D crystals and are expected to have quantized anomalous Hall conductivities with Chern numbers $(C)$ that increase by one when the film thickness increases by $\triangle t \sim \pi/k_w$ [18]. For the case of Sb doping $x \sim 25\%$ as illustrated by the red curves in Fig. 1a, b, the position of the Weyl point $(k_w \approx 3\pi/25d)$ shifts closer to the $\Gamma$ point than at $x = 0$ case, and thus corresponds to the expansion of the $C = 1$ state to larger film thicknesses. Figure 1b shows the theoretical thin film Chern gaps versus thickness obtained by fitting the bulk DFT bands to a simplified model [18] as detailed in the Supplemental Note 1. The gaps close when a topological phase transition occurs between the different Chern numbers. As the Weyl semimetal state can exist only under the spin-aligned condition, our focus is thus on the Chern insulator states in the magnetic field induced spin-moment alignment phase, which can be observed more consistently in $Mn(Bi_{1-x}Sb_x)_2Te_4$ thin films.

## Electrical transport in $Mn(Bi_{1-x}Sb_x)_2Te_4$ films

We first examine the low-temperature transport properties of the $Mn(Bi_{1-x}Sb_x)_2Te_4$ films. Figure 1c plots the four-terminal resistivity $(\rho_{xx}^{CNP})$ measured at the CNP as a function of $Mn(Bi_{1-x}Sb_x)_2Te_4$ thickness for different Sb substitution levels ($x = 0$, 0.20, and 0.26). We refer to a sample as surface-like when its conductivity is thermally-activated at low temperatures, suggesting the presence of a bulk energy gap. Samples classified as bulk-like have weaker temperature dependence and are presumed to have disorder-induced bulk states at all energies.

The representative resistivity versus temperature curves for the $Mn(Bi_{1-x}Sb_x)_2Te_4$ are included in Supplementary Figs. 5, 6 to distinguish the surface-like and bulk-like behaviors. The surface-like $\rho_{xx}^{CNP}$ behavior persists to the largest thickness range for the Sb concentration $x = 0.26$ at $T = 2$ K while $\rho_{xx}^{CNP}$ is bulk-like for thickness above 21-SLs. For $x = 0.20$, $\rho_{xx}^{CNP}$ decreases abruptly at thicknesses above 12-SLs. This trend persists for $MnBi_2Te_4$. This trend is expected since $MnBi_2Te_4$ has bulk n-type doping [5,24], while the substitution of Sb on the Bi sites can shift the Fermi level of the bulk band toward p-type doping, with $x = 0.26$ being closest to CNP [20]. Sb doping at higher concentrations $x > 0.26$ leads to excessive p-type doping and prevents access to CNP in thin flakes [21]. Moving the Fermi level by the Sb to Bi ratio can thus maximize the surface-like regime for probing their quantum transport properties.

Figure 1d shows a representative $\rho_{xx}$ curve as a function of back-gate voltage $(V_{bg})$ for an 18-SL $Mn(Bi_{0.74}Sb_{0.26})_2Te_4$ device measured at a temperature of 2 K. The ambipolar gate-dependent $\rho_{xx}$ suggests a bulk gap with an intrinsic surface state at the CNP ($V_{bg} \sim +3$ V). The surface carrier density can be tuned to either hole or electron transport by controlling the gate voltage. The field-effect mobility $\mu_{FE} = \frac{1}{C_g} \frac{dG_{xx}}{dV_g}$, where $C_g$ is the gate capacitance ($\approx 80$ nF/cm² for a ~ 30 nm mica dielectric), $G_{xx}$ ($= 1/\rho_{xx}$) is the four-terminal conductance, and $V_g$ is the voltage applied through the graphite gate-electrode, is plotted in Fig. 1d. We see that the electron mobility increases with gate voltage (electron density), while the hole mobility responds weakly and remains small in the low gate voltage (hole density) regime. The mobility is more than one order of magnitude higher mobility for electrons compared to hole carriers. By applying dual-gate voltages, the top and bottom surface carrier densities can be modulated to

achieve electron mobilities as high as 4000 cm²/Vs at a total carrier density of $>5 \times 10^{11}$ cm².

## $C = 1$ state in Mn(Bi$_{1-x}$Sb$_x$)$_2$Te$_4$ films

The magnetic field-dependent transport properties of Mn(Bi$_{1-x}$Sb$_x$)$_2$Te$_4$ films with the Sb concentrations $x = 0$, 0.20, and 0.26 for a variety of thicknesses were studied. When a perpendicular magnetic field is applied, Mn(Bi$_{1-x}$Sb$_x$)$_2$Te$_4$ undergoes the spin–flop and spin–flip transitions[20,22] from the antiferromagnetic (AFM) to the canted, and finally to the aligned spin-moment configurations. To compare the magnetic field dependence of the Mn(Bi$_{1-x}$Sb$_x$)$_2$Te$_4$ film with different Sb substitutions, we plot the color maps of $\rho_{yx}$ as functions of gate voltage and magnetic field for the 8-SL MnBi$_2$Te$_4$, 10-SL Mn(Bi$_{0.8}$Sb$_{0.2}$)$_2$Te$_4$, and 21-SL Mn(Bi$_{0.74}$Sb$_{0.26}$)$_2$Te$_4$, respectively, in Fig. 2a–c. The spin-flop and spin-flip transition fields, as determined from the kinks in their $\rho_{xx}$ and $\rho_{yx}$ versus magnetic field curves, happen at magnetic fields of ~±2–3 T and ~±7 T, respectively, are observed in all the samples. The magnetic transition fields identified by the color line marks depicted in Fig. 2a–c agree with their parent bulk compounds[20] and theoretically calculated values[25]. Line profiles of $\rho_{xx}$ and $\rho_{yx}$ curves for the 8-SL MnBi$_2$Te$_4$, 10-SL Mn(Bi$_{0.8}$Sb$_{0.2}$)$_2$Te$_4$, and 21-SL Mn(Bi$_{0.74}$Sb$_{0.26}$)$_2$Te$_4$ are plotted in Fig. 2d–f. The $\rho_{yx}$ increases

sharply with magnetic field in the canted antiferromagnetic (CAFM) phase, and saturates at ~h/e² as the thin film is driven into the FM phase by magnetic field. The suppression of $\rho_{xx}$ in the FM phase further confirms the development of $C = 1$ state in all three samples. The interpretation is supported by the gate-dependent $\rho_{xx}$ and $\rho_{yx}$ curves measured at magnetic field of 9 T for the three respective samples in Supplementary Figs. 7a, 8, and 10, respectively, where the $\rho_{yx}$ plateau and $\rho_{xx}$ minimum can be seen. Despite not being fully quantized, the 21-SL Mn(Bi$_{0.74}$Sb$_{0.26}$)$_2$Te$_4$ film exhibits all the features of the $C = 1$ state. The anomalous Hall loop at low magnetic field for the 8-SL MnBi$_2$Te$_4$ and 10-SL Mn(Bi$_{0.8}$Sb$_{0.2}$)$_2$Te$_4$ could be due to the uncompensated surface magnetization[11] or antiferromagnetic domain walls[26], whereas no zero field hysteretic behavior observed in the 21-SL Mn(Bi$_{0.74}$Sb$_{0.26}$)$_2$Te$_4$ film. The observed $C = 1$ states in all three Mn(Bi$_{1-x}$Sb$_x$)$_2$Te$_4$ films with the different Sb substitutions confirm that their spin-alignment configuration is in the topological phase with a Chern insulator gap.

To further evaluate the thickness-dependence of the $C = 1$ state, we plot $\rho_{xx}$ and $\rho_{yx}$ values as a function of thickness, with the gate voltage tuned to the $\rho_{yx}$ maximum for each thickness, at magnetic field of 9 T. Our primary observation is that, despite their similar field-dependent quantization behavior, the $C = 1$ state prolongs to the larger

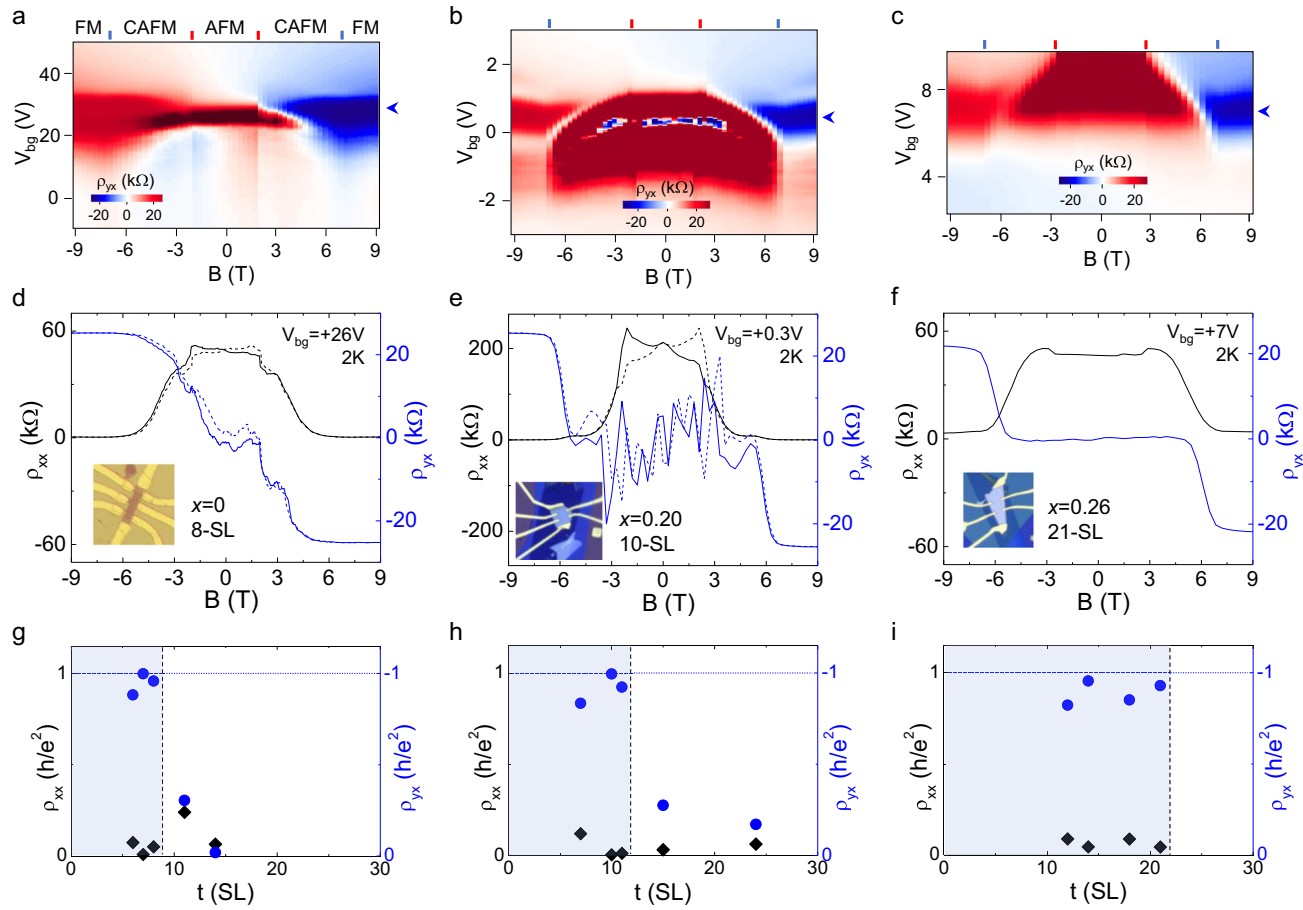

**Fig. 2 | Magnetic field induced quantization at different Sb substitution levels.** Color maps of $\rho_{yx}$ as a function of magnetic field and gate voltage for (**a**) MnBi$_2$Te$_4$, (**b**) Mn(Bi$_{0.8}$Sb$_{0.2}$)$_2$Te$_4$, and (**c**) Mn(Bi$_{0.74}$Sb$_{0.26}$)$_2$Te$_4$ devices at flake thickness of 8-SLs, 10-SLs, and 21-SLs, respectively, measured at temperature of 2 K. The red and blue lines mark the spin–flop and spin–flip transition fields, respectively, for each sample. The AFM, CAFM, and FM denote the antiferromagnetic, canted antiferromagnetic, and ferromagnetic phases, respectively. The $\rho_{xx}$ and $\rho_{yx}$ line profiles in (**d**), (**e**), and (**f**) as a function of magnetic field are extracted from the color maps for the respective devices in (**a**), (**b**), and (**c**) at the gate voltages indicated by the

blue arrows. The respective device images are inserted in (**d**–**f**). The color maps present the raw data without antisymmetrization. The $\rho_{xx}$ and $\rho_{yx}$ line profiles in (**d**)–(**f**) are symmetrized and antisymmetrized, respectively, with respect to the magnetic field. The high resistive $\rho_{yx}$ (truncated region) at low magnetic field in (**a**)–(**c**) is due to the mixing from the magnetoresistance. The extracted $\rho_{xx}$ (black rhombus) and $\rho_{yx}$ (blue circle) values at the maximum $\rho_{yx}$ as a function of flake thickness for (**g**) MnBi$_2$Te$_4$, (**h**) Mn(Bi$_{0.8}$Sb$_{0.2}$)$_2$Te$_4$, and (**i**) Mn(Bi$_{0.74}$Sb$_{0.26}$)$_2$Te$_4$ devices measured at magnetic field of 9 T. The blue color shades in (**g**)–(**i**) denote the thickness range where the $C = 1$ state is observed in the spin-aligned state.

thickness range with the Sb substitutions as shown in Fig. 2g–i. This trend is consistent with our DFT calculations, which indicate the shift of the Weyl point position by Sb doping. In Fig. 2g, the observation of $C = 1$ quantum Hall states up to 8-SL $MnBi_2Te_4$ agrees with our calculations (Fig. 1b) and the literature[10]. The thickness limit for the $C = 1$ state extends to 11-SLs for the $Mn(Bi_{0.8}Sb_{0.2})_2Te_4$ as shown in Fig. 2h. The 16-SL $Mn(Bi_{0.8}Sb_{0.2})_2Te_4$ (Supplementary Fig. 9) shows the absence of $C = 1$ state at magnetic field up to 18 T, presumably due to the excessive bulk conduction channels in this sample. In Fig. 2i, we show the substantially wider thickness range of the $C = 1$ state resolved for the $Mn(Bi_{0.74}Sb_{0.26})_2Te_4$. Although this can be somehow related to the extension of the surface-like regime in thin films by the optimal Sb substitutions, according to our calculations in Fig. 1b, the $Mn(Bi_{0.74}Sb_{0.26})_2Te_4$ films at the given film thickness range should lie in the higher Chern number states.

## Dual-gate tuning of Chern states

To further identify the Chern insulator states in our thicker $Mn(Bi_{0.74}Sb_{0.26})_2Te_4$, we perform a detailed analysis for these devices in a dual-gating platform. Figure 3a–f compare the dual-gate maps of longitudinal conductivity ($\sigma_{xx}$) and Hall resistivity ($\rho_{yx}$) for $Mn(Bi_{0.74}Sb_{0.26})_2Te_4$ devices at the thickness of 21, 18, and 14-SLs, respectively, measured at 9 T. As shown in Fig. 3a and b, the 21-SL

$Mn(Bi_{0.74}Sb_{0.26})_2Te_4$ reveal a clear $C = 1$ plateau in the dual-gate maps. This is also indicated by the $\sigma_{xy}$ ($\sigma_{xx}$) versus $V_{bg}$ line profiles, as shown in Fig. 3g. In this device, we observe no other Chern states develop near the CNP besides the $C = 1$ state at the highest accessible magnetic field of 9 T. Careful tracking of the $C = 1$ state in magnetic field reveals its formation at Fermi energy slightly below the CNP as detailed in Supplementary Figs. 10 and 11. Similar behavior was also observed in the 18-SL $Mn(Bi_{0.74}Sb_{0.26})_2Te_4$ (Supplementary Figs. 12, 13).

The 18-SL $Mn(Bi_{0.74}Sb_{0.26})_2Te_4$ device with higher electron mobility (Fig. 1d) shows more quantization features at 9 T. With the dual-gating structure, we can access these quantum Hall states by tracing the boundaries of these states with dashed lines in the dual-gate maps. The color map of $\sigma_{xx}$ versus dual-gate voltages depicted in Fig. 3c shows multiple minima corresponding to the different quantized $\rho_{yx}$ plateaus as indexed in the dual-gate map in Fig. 3d, showing the different Chern numbers. The $\sigma_{xy}$ ($\sigma_{xx}$) versus $V_{bg}$ line profiles depicted in Fig. 3h reveal the well-developed $C = 3$ plateau and the developing $C = 1$ and 5 quantization states. Tracking the Chern state development at the lower magnetic field reveals an additional $C = 2$ state in the CAFM phase (Supplementary Fig. 14). The existence of the $C = 2$ state in the CAFM phase is also verified by the $\rho_{xx}$ and $\rho_{yx}$ dual-gate maps and the flow diagram in ($\sigma_{xx}$, $\sigma_{xy}$) parameter space swept at magnetic field of 6 T as shown in Supplementary Figs. 12, 14,

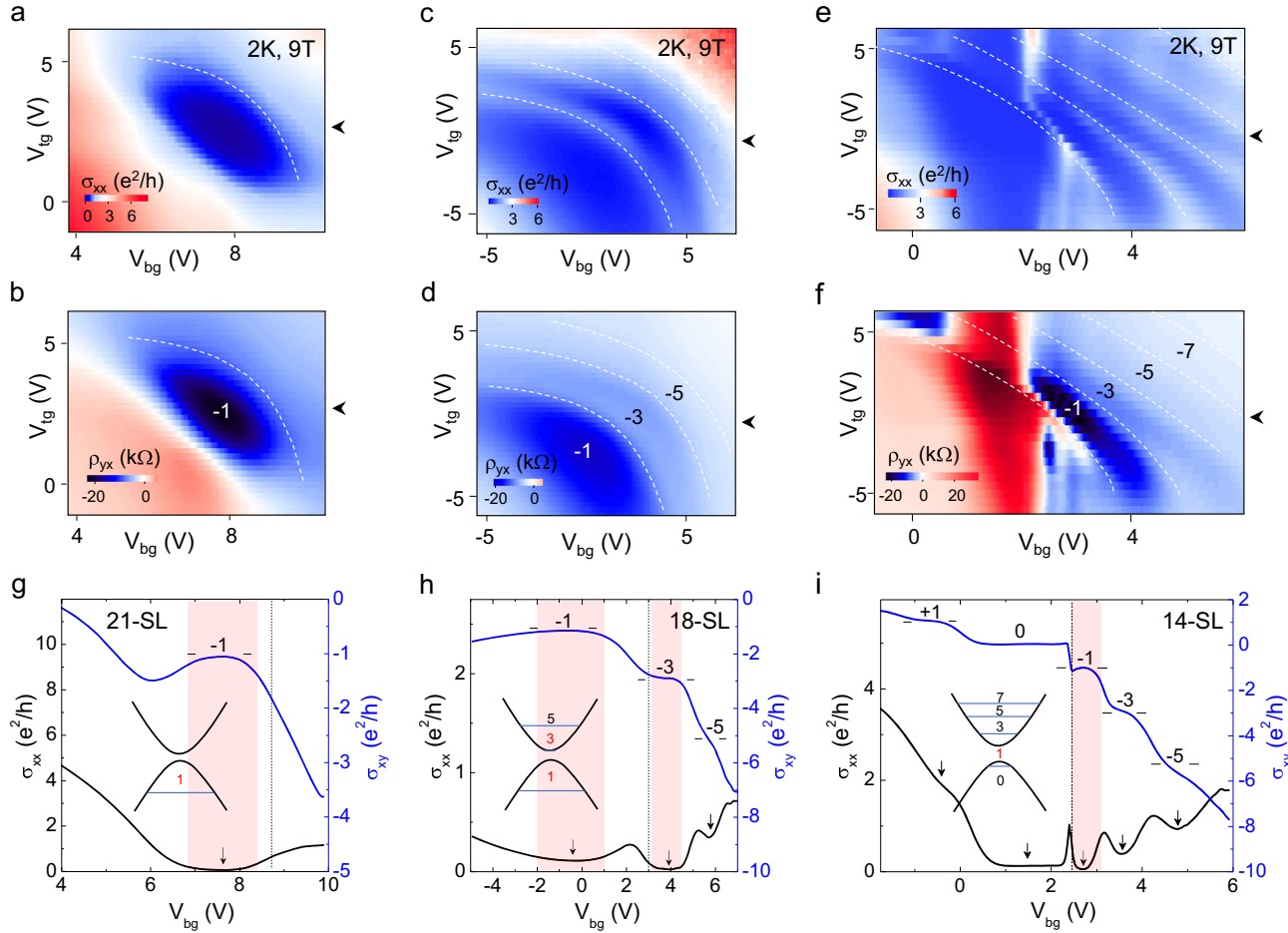

**Fig. 3 | Tunable Chern states by dual-gating.** Color maps of $\sigma_{xx}$ and $\rho_{yx}$ as a function of dual-gate voltages for $Mn(Bi_{0.74}Sb_{0.26})_2Te_4$ at flake thicknesses of (**a, b**) 21-SLs, (**c, d**) 18-SLs, and (**e, f**) 14-SLs, respectively, measured at temperature of 2 K and magnetic field of 9 T. The white dashed lines in the color maps trace the boundaries of the quantization plateaus with the respective quantum states indexed in the $\rho_{yx}$ maps. Line profiles of $\sigma_{xx}$ and $\sigma_{xy}$ versus backgate voltage curves swept across the charge neutrality as indicated by the black arrows in the color

maps for the (**g**) 21-SLs, (**h**) 18-SLs, and (**i**) 14-SLs $Mn(Bi_{0.74}Sb_{0.26})_2Te_4$. The $\sigma_{xy}$ plateaus and the corresponding $\sigma_{xx}$ minima (indicated by the black arrows) are indexed to the Chern numbers in (**g**)–(**i**). Vertical dashed lines in (**g**)–(**i**) mark the backgate voltages corresponding to the $\rho_{xx}^{CNP}$ determined at zero magnetic field. The surface band structures in (**g**)–(**i**) are sketched to illustrate the LL spectra observed for the respective thicknesses at 9 T. The red shades in (**g**)–(**i**) denote the Chern states forming near the CNP.

respectively. Also, we note that the $C = 2$ state coincides with the $\rho_{xx}^{CNP}$ at the zero magnetic field.

We further analyze the quantization states in the 14-SL Mn(Bi$_{0.74}$Sb0.26)$_2$Te$_4$. In addition to the $C = 1$ state, the dual-gate maps in Fig. 3e, f show a series of oscillatory $\sigma_{xx}$ minima and $\rho_{yx}$ plateaus, respectively, corresponding to the different quantum Hall states develop at 9 T when tuning the dual-gate voltages. The linecuts of $\sigma_{xx}$ and $\sigma_{xy}$ as a function of backgate voltage at magnetic field of 9 T, as depicted in Fig. 3i, reveal the quantum Hall plateaus with $C = 0, 1, 3, 5,$ etc. The color maps of $\rho_{xx}$ and $\rho_{yx}$ as functions of magnetic field and backgate voltage (Supplementary Fig. 15) resolve the fan diagram of Landau levels at odd integer fillings, where the $\rho_{xx}$ minima for each filling factor can be traced down linearly to a gate voltage at zero magnetic field corresponding to the CNP. Such a feature is similar to the surface states' LL fan diagram in non-magnetic topological insulators[27,28]. Different from the 18-SL and 21-SL Mn(Bi$_{0.74}$Sb$_{0.26}$)$_2$Te$_4$, the $\rho_{xx}^{CNP}$ obtained at zero magnetic field sits between $C = 0$ and 1 states at high magnetic field for the 14-SL Mn(Bi$_{0.74}$Sb$_{0.26}$)$_2$Te$_4$ (Supplementary Fig. 16). The schematic diagrams inserted in Fig. 3g–i illustrates the surface band structures with different Chern states resolved in the respective thicknesses of the Mn(Bi$_{0.74}$Sb$_{0.26}$)$_2$Te$_4$ films.

## Anomalous Landau levels in magnetic topological insulators

To interpret the rich Chern insulator states observed in our Mn(Bi$_{0.74}$Sb$_{0.26}$)$_2$Te$_4$ films, we calculate their LLs spectra using a simplified model[18], in which the 2D massive Dirac cones are coupled by

tunneling within and between the compound's septuple-layer building blocks. For explanation purposes, we first present in Fig. 4a the LLs spectrum for a non-magnetic TI thin film[29] which has no exchange coupling. The degenerate $n = 0$ LLs (red curves) lie at the Fermi level due to the non-trivial Berry's phase[30] in addition to the typical $n \neq 0$ bulk LLs (black curves). When the exchange coupling is introduced, the quasi-2D films have discrete finite-length-chain hopping eigenstates, which have no spin-orbit coupling but are spin-split by exchange interactions with the aligned Mn spins. A series of spin-polarized anomalous LLs (red and blue lines) with magnetic field-independent energies emerge from the $\vec{k}_\perp = 0$ states, as shown in Fig. 4c, d. The derivation of anomalous LLs bands from a generalized Su–Schrieffer–Heeger (SSH) model (Supplementary Note 1) and their wavefunction distributions can be found in Supplementary Fig. 1. The Chern numbers of the anomalous LLs are determined by $(N_{E<E_F} - N_{E>E_F})/2$ where $N_{E<E_F}$ ($N_{E>E_F}$) is the total number of subbands below (above) Fermi level. Noted that for convenience in calculations, we ignore the magnetic transitions and assume the spin-aligned phase for all magnetic field ranges. This assumption enables the determination of the actual $C$ due to the band topology in zero magnetic field and the development of anomalous LLs under high magnetic field. For example, the $C = 2$ can be identified from the LLs spectra in Fig. 4c, d by tracing down to the zero magnetic field limit. The $C = 2$ can be further verified by the $C$ calculated from the bulk Hall conductivity at the same layer thickness of the Mn(Bi$_{0.75}$Sb$_{0.25}$)$_2$Te$_4$ films (Fig. 1b). Under high magnetic field, the non-anomalous LLs with n $\neq$ 0 indices (black curves in Fig. 4c, d) in conduction-valence pairs move further away from the Fermi level as magnetic fields strengthen. This leaves an

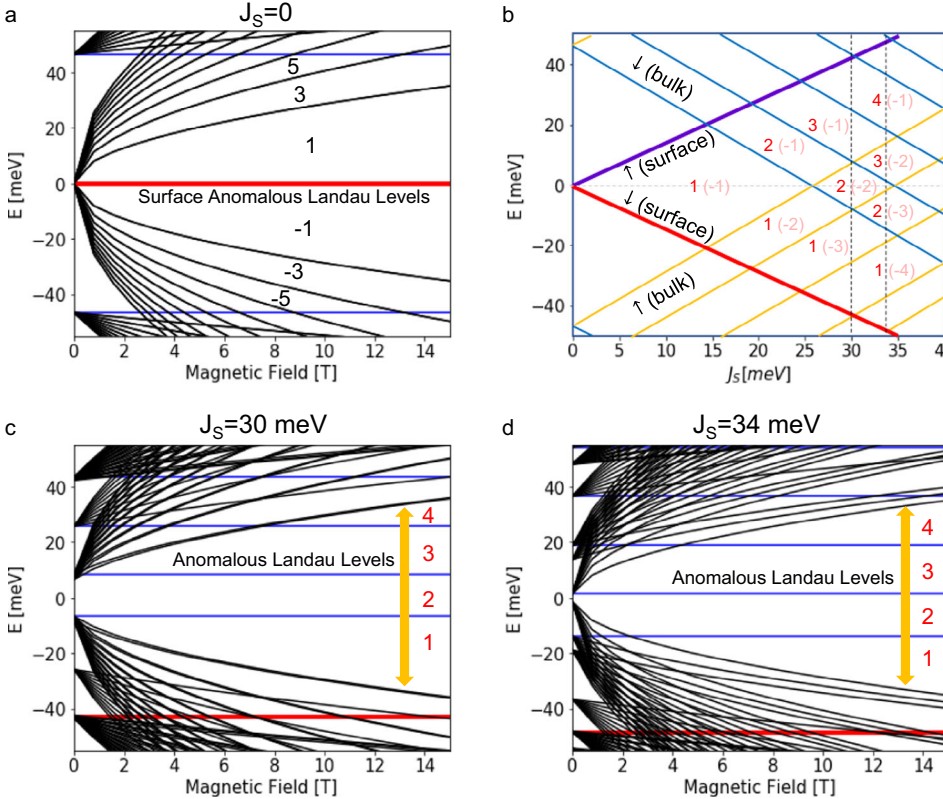

**Fig. 4 | Spectra of anomalous and non-anomalous Landau levels.** Landau level fan diagrams and filling factors of an 18-SL Mn(Bi$_{0.75}$Sb$_{0.25}$)$_2$Te$_4$ film. **a** The Landau level structures for the case of a non-magnetic thin film ($J_S = 0$). The $n \neq 0$ non-anomalous Landau levels are plotted with black curves, while the $n = 0$ anomalous Landau levels whose energies are independent of magnetic field are plotted with blue and red curves. The red curves distinguish anomalous Landau levels that are localized at the surface. **b** Band energies at 2D wavevector **k** = 0 at zero magnetic field versus the same-layer exchange splitting, $J_S$. Spin up (down) states distributed

in the bulk are labeled with orange (blue) color, and the bold purple (red) curve is for the spin up (down) state localized at the thin film surfaces. The Chern numbers of the anomalous LLs for down (up)-spins are indexed in (**b**). **c, d** Landau level structures for thin film with an aligned moment spin configuration at $J_S = 30$ and 34 meV, respectively, labeled in (**b**) with black vertical dashed lines. Strong quantum Hall states occur when only anomalous Landau levels are close the Fermi level. The orange arrows in (**c**) and (**d**) denote the region where the anomalous Landau levels developed at high magnetic fields.

interval of carrier density in which only the $n = 0$ anomalous LLs (red and blue lines in Fig. 4c, d) are present close to the Fermi level as indicated by the yellow arrows, and the gaps between these levels are large enough to support Hall quantization that is robust against disorder. The energy gaps of the anomalous LLs depend on the $n \neq 0$ non-anomalous LLs and thus are magnetic field dependence.

A general picture of the relationship between the Chern numbers and exchange coupling ($J_S$) is illustrated in Fig. 4b where the surface and bulk spin-splitting bands are plotted. In the case of $J_S = 0$, the number of subbands above and below the Fermi level equals ($N_{E<E_F} = N_{E>E_F}$), and thus the Chern number is 0. When the exchange field is turned on ($J_S \neq 0$), the anomalous LLs are spin-polarized with two nearly degenerate surface anomalous LLs labeled represented by the green and red curves in Fig. 4b. The filling factor at the Fermi level becomes 1 (−1) for down-spin (up-spin) as the $N_{E<E_F}$ and $N_{E>E_F}$ are now differed by one. When the crossings between up-spin (yellow curves) and down-spin (blue curves) happen at the Fermi level by further increasing $J_S$, the filling factor will increase by one. Subsequently, the Chern quantization is therefore expected to be observable over a range of filling factors magnitudes centered on 1+ the number of $\vec{k}_\perp = 0$ crossings that occur between up and down spins as indexed in Fig. 4b. When the magnetic field is reversed, both the spin of the anomalous LLs and the sign of the exchange coupling between the local moments and the Dirac electrons are reversed, therefore the sign of the filling factor will follow the sign of the magnetic field. The dependency between the spin of the anomalous LLs and the sign of the exchange coupling further infers that the anomalous LLs are spin-polarized.

We also note that since the position of Weyl points $k_w$ depends on the exchange splitting[18], which may be reduced by antisite defects[31] leading to a change in the Chern numbers and the gaps in zero magnetic field spin alignment phase (Supplementary Fig. 2). To further elaborate on this effect, we examine two cases of $J_S = 30$ and 34 meV for the 18-SL Mn(Bi$_{0.75}$Sb$_{0.25}$)$_2$Te$_4$ film in Fig. 4c, d, respectively. As shown in their LL spectra, the $J_S = 30$ and 34 meV exhibit very different $C = 2$ Chern insulator gaps at zero magnetic field. Nevertheless, the well-spaced anomalous LLs dominate over a finite region of filling factor at high magnetic field does not vary too much with the $J_S$.

Finally, we compare our experimental results for the Mn(Bi$_{0.74}$Sb$_{0.26}$)$_2$Te$_4$ films with the calculated LLs structure at a similar thickness and Sb doping level. We plot in Supplementary Fig. 17 the calculated LL gaps at different filling factors as a function of magnetic field for the Mn(Bi$_{0.75}$Sb$_{0.25}$)$_2$Te$_4$ films at different thicknesses. The LL gap size of each filling factor is determined either by gaps between anomalous LLs or by gaps between anomalous ($n = 0$) and non-anomalous ($n \neq 0$) LLs. Under a strong magnetic field, the calculated filling factor of $C = 1$ exhibits the largest LL gap in all three thicknesses of Mn(Bi$_{0.75}$Sb$_{0.25}$)$_2$Te$_4$ film. This explains the experimental observation where the $C = 1$ state was observed over the wide thickness range in our Mn(Bi$_{0.74}$Sb$_{0.26}$)$_2$Te$_4$ film. However, one noticeable feature in the case of larger film thickness is that the $C = 1$ filling shifts below the Fermi level and the surface gap at the Fermi level change to the filling factor of higher Chern number $C = 2$ (refer to Supplementary Figs. 4 and 17 for details). This is consistent with the observation in our 18-SL Mn(Bi$_{0.74}$Sb$_{0.26}$)$_2$Te$_4$ film, indicating that the 18-SL is in a higher Chern number $C = 2$ state. Moreover, the calculated gap size for the $C = 3$ state exceeds the $C = 2$ gap at high magnetic field, which explains the well-developed $C = 3$ state at 9 T at this film thickness.

Whereas for 14-SL Mn(Bi$_{0.74}$Sb$_{0.26}$)$_2$Te$_4$ film, both of our experimental observation and calculations suggest that the 14-SL Mn(Bi$_{0.74}$Sb$_{0.26}$)$_2$Te$_4$ film in the $C = 1$ state (Supplementary Figs. 3 and 17) with the higher filling factors of $C = 3$ and 5 states assigned to the non-anomalous band LLs. The calculations infer an anomalous LLs $C = 2$ state can develop at the higher magnetic field. To verify this, we performed measurements at a high magnetic field up to 18 T for the 11-SL and 10-SL Mn(Bi$_{0.8}$Sb$_{0.2}$)$_2$Te$_4$ films, both with the $C = 1$ state at the

Fermi level, as depicted in Supplementary Fig. 18. Similar to the 14-SL Mn(Bi$_{0.74}$Sb$_{0.26}$)$_2$Te$_4$ film, the quantization steps of $C = 0$, 1, and 3 states can be observed in the 11-SL and 10-SL Mn(Bi$_{0.8}$Sb$_{0.2}$)$_2$Te$_4$ at a magnetic field of 10 T. While the additional $C = 2$ plateau starts to develop at higher magnetic field of >14 T in both samples. The phase boundaries traced by the red dashed lines in Supplementary Fig. 18(b) and (e) covering the $C = 1$ and 2 plateaus are consistent with our theoretical picture of the Chern insulator $C = 1$ state and anomalous LLs $C = 2$ forming at high magnetic field as the ordinary $n \neq 0$ bulk band LLs of $C = 3$ and above states move away from Fermi level. Our results can thus support the existence of the spin-polarized anomalous LLs in Mn(Bi$_{1-x}$Sb$_x$)$_2$Te$_4$ thin film Weyl semimetals, which emerges at larger film thickness or strong magnetic field near the Fermi level.

## Discussion

In summary, we studied the magnetoelectrical transport of the intrinsic MTI Mn(Bi$_{1-x}$Sb$_x$)$_2$Te$_4$ for $x = 0$, 0.20, and 0.26 by probing Chern quantization states and their relationship with the flake thickness and Sb concentrations. We identified the thickness ranges for surface-like insulating and bulk-like metallic transport regimes. The thickness-dependent Hall conductivities, particularly for the $C = 1$ Chern insulator state in the Mn(Bi$_{1-x}$Sb$_x$)$_2$Te$_4$, show a correlation with the separation of Weyl points as described by our theoretical models, indicating that Mn(Bi$_{1-x}$Sb$_x$)$_2$Te$_4$ behaves as thin film Weyl semimetals. Our transport results for different Sb concentrations highlight the importance of the Weyl point separation in the spin-aligned magnetic phase to the Hall quantization. Moreover, we showed that the Mn(Bi$_{1-x}$Sb$_x$)$_2$Te$_4$ at larger film thickness and strong magnetic field can give rise to the unusual quantization sequence and the intriguing anomalous LLs near the Fermi level. Our work illustrates the complexity of the intertwined topological surface states and ferromagnetism in Landau quantization and thus can serve as a guide to bridge the gap between the 2D Chern insulators and 3D Weyl semimetals.

## Methods

### Materials

Mn(Bi$_{1-x}$Sb$_x$)$_2$Te$_4$ bulk crystals at different Sb doping levels were grown by a self-flux growth method[20,24]. Variable thicknesses of Mn(Bi$_{1-x}$Sb$_x$)$_2$Te$_4$ thin flakes were exfoliated from the parent bulk crystals and then transferred into the heterostructures of graphite/muscovite mica sandwiched layers using a micromanipulator transfer stage. The graphite and muscovite mica layers were subsequently transferred onto Si/SiO$_2$ substrate using polypropylene carbonate, and followed by an annealing process in argon gas to clean the polymer residues. The graphite/muscovite mica layers serve as the gate-electrode/dielectric layers. The Mn(Bi$_{1-x}$Sb$_x$)$_2$Te$_4$ exfoliation and transfer processes were performed in an argon gas-filled glovebox with O$_2$ and H$_2$O levels <1 ppm and <0.1 ppm, respectively, to prevent oxidation in thin flakes. We fabricated the Mn(Bi$_{1-x}$Sb$_x$)$_2$Te$_4$ devices into the Hall bar configuration using a standard electron beam lithography process and metal deposition with Cr/Au (20 nm/60 nm) as the contact electrodes using a CHA Solution electron beam evaporator. The Mn(Bi$_{1-x}$Sb$_x$)$_2$Te$_4$ flakes were protected by polymethyl methacrylate (PMMA) while transporting for lithography and metal deposition processes.

### Measurements

Low-temperature magnetotransport measurements were performed in a Quantum Design Physical Properties Measurement System (PPMS) in helium-4 circulation (2–300K) and magnetic field up to 9 T. Two synchronized Stanford Research SR830 lock-in amplifiers at a frequency of 5–8 Hz were used to measure the longitudinal and Hall resistances concurrently on the Mn(Bi$_{1-x}$Sb$_x$)$_2$Te$_4$ devices. The devices were typically sourced with a small AC excitation current of

20–100 nA. Two Keithley 2400 source meters were utilized to source DC gate voltages separately to the top and bottom gate electrodes. Magnetotransport measurements at high magnetic field were carried out in a helium-3 variable temperature insert at a base temperature of 0.4 K and magnetic field up to 18 T based at the National High Magnetic Field laboratory.

## Theoretical calculations

DFT calculations were performed using Vienna Ab initio Simulation Package (VASP)[32] in which Generalized gradient approximations (GGA) of Perdew-Burke-Ernzerhof (PBE)[33] have been adopted for exchange-correlation potential. On-site correlation on the Mn-3d states is treated by performing DFT + U calculations[34] with U−J as 5.34 eV. The global break condition for the electronic SC-loop is set to be $10^{-7}$ eV and the cutoff energy for the plane wave basis set is 600 eV during the self-consistent (SC) calculations. A $9 \times 9 \times 6$ Gamma-centered k-point integration grid was employed with Gaussian broadening factors as 50 meV. In the calculations of bulk Mn$(Bi_{1−x}Sb_x)_2Te_4$, supercells of $2 \times 2 \times 1$ unit cell were used to model the doping density of Sb atoms with densities of 0, 25%, 50%, 75%, and 100%. The calculations of Landau levels are based on the coupled Dirac cone model illustrated in the supplemental material, with the parameters estimated from the DFT calculations.

## Data availability

The data supporting the findings of this study are available within the article and supplementary information. The main data generated in this study are publicly available at https://doi.org/10.5068/D1097T. Additional data are available from the corresponding authors upon request.

## Code availability

The codes used for the numerical simulation are available from the corresponding authors upon request.

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

## Acknowledgements

This work was supported by the National Science Foundation the Quantum Leap Big Idea under Grant No. 1936383 and the U.S. Army Research Office MURI program under Grants No. W911NF-20-2-0166 and No. W911NF-16-1-0472. Support for crystal growth and characterization was provided by the National Science Foundation through the Penn

State 2D Crystal Consortium-Materials Innovation Platform (2DCC-MIP) under NSF cooperative agreement DMR-2039351. A portion of this work was performed at the National High Magnetic Field Laboratory, which is supported by the National Science Foundation Cooperative Agreement No. DMR-1644779 and the State of Florida.

## Author contributions

S.K.C. and K.L.W. planned the experimental project. S.H.L. and Z.M. prepared the bulk crystals. S.K.C. fabricated the devices and conducted the transport measurements. J.J. helped with transport measurements conducted at National High Magnetic Field laboratory. C.L. and A.H.M. performed the theoretical calculations. S.K.C., L.C., A.H.M., and K.L.W. wrote the manuscript. All authors discussed the results and commented on the manuscript.

## Competing interests

The authors declare no competing interests.
