## [Peer Review File · Nature Communications]

REVIEWER COMMENTS

Reviewer #1 (Remarks to the Author):

This work discusses the interplay between the intrinsic Chern insulator state in $\text{MnBi}_{2-x}\text{Sb}_x\text{Te}_4$ with 3 different compositions and Landau quantization. Their conclusions seem 2-fold, first the thickness range over which the Chern insulator state is observed is extended to large thicknesses for high Sb content due to the changing Weyl node spacing. Second, an interplay between the Chern insulator state and Landau quantization gives rise to a set of anomalous Landau levels with an unconventional sequence of quantization index (Chern number). This work has the potential to be impactful and noteworthy but, in some instances, the main message of the manuscript is not clearly presented and diluted in detail. Because of this, I cannot yet tell if the second conclusion pertaining to the interpretation of the Landau quantized regime is well supported by data. Can the authors address the following issues:

- To better address the issue that I'm facing, can the authors construct a Landau fan diagram similar to the ones shown in Fig. 4 for each sample shown in Fig. 3?

- The thickness of the flakes shown in Fig. 3 is somewhat large compared to what is typically studied in the literature, and raises the question of whether the two gates are actually able to deplete the bulk (especially in the case of the 21-SL sample). Is it possible that in this sample the authors are controlling the amount of band bending at the top and bottom surface?

- Line 91: Is the gate dielectric made out of mica or hBN as described in the methods section. If it's mica please describe the process by which mica was transferred in the methods section.

- The theory predicts a specific behavior as a function of positive versus negative magnetic field: "When the magnetic field is reversed, the spin of the anomalous LLs is reversed, but the sign of the exchange coupling is also reversed, so the sign of the filling factor range at which strong quantum Hall effects [sic] will not change." Is that consistent with the data? Can the authors show data where this is compared?

- The authors claim to observe a signature of the $C=1$ filling shifting below the Fermi level and that the surface gap at the Fermi level is at filling factor of higher Chern number, but how can they specifically determine the position of the surface gap in their measurement? For instance, in Fig. 3i, both $C=0$ and $C=1$ overlap with the minimum in conductivity, making the precise position of the surface gap (or the charge neutrality point) ambiguous versus gate voltage.

- Line 231 to 237: the authors compare their findings in the 18-SL sample from Fig. 3 with theory, referring to Fig. S3, but Fig. S3 does not include a calculation for 18-SLs.

- I am having trouble deciphering the color maps in the Fig. 2. For instance in Fig. 2(c) does the dark red +/- 6T correspond to a region with very high Hall resistivity or zero Hall resistivity? Or is that showing mixing with the magnetoresistance?

- The Hall resistivity plateau in Fig. 2f is very wide, and seems robust up to more than 4T. This behavior seems inconsistent with the magnetic phase boundaries shown in Fig. 2c. Is this behavior understood?

- There needs to be a more intuitive explanation of the meaning of "anomalous Landau levels" and how they determine the value of the Chern number. Is it due to a changing number of inverted bands driven by spin-splitting or is it something more complex?

- Why are $J_s=30\text{meV}/34\text{meV}$ chosen to compare with data?

Reviewer #2 (Remarks to the Author):

MnBi₂Te₄ and its family of materials have brought great progress in studying novel topological quantum states. Chong et al. study the Chern insulator states in the intrinsic magnetic topological material Mn(Bi_{1-x}Sb_x)₂Te₄ devices and their relationship with the device thickness and Sb concentrations. Firstly, they find that the thickness range for the C= 1 state shows a correlation with the Sb concentrations and they attribute this behavior to the change of separation of Weyl points induced by Sb substitution. Then, their transport results for different Sb concentrations suggest the existence of anomalous Landau levels in Mn(Bi_{1-x}Sb_x)₂Te₄, which are attributed to the interplay of B=0 Chern insulator state and Landau quantization. These interesting results indicate that Mn(Bi_{1-x}Sb_x)₂Te₄ devices may provide a tunable platform for probing the physics of new topological states, which may stimulate further theoretical and experimental exploration of nontrivial topology in MnBi₂Te₄ and its family of materials. Overall I recommend publication, provided that the authors address the following questions:

1. The authors attribute the anomalous Landau Levels to the interplay of B=0 Chern insulator state and Landau quantization. However, no Chern insulator state can be experimentally observed under B=0 in this paper. How to understand the so-called B=0 Chern insulator state theoretically?

2. Some key statements in this paper are improper, unclear and puzzling. For example, in Line 44, the authors state that “We show that Sb substitution extends the surface gap regime to a wider thickness range by suppressing conduction from the trivial bulk bands, enabling better access to their Weyl physics”, which is puzzling since the surface gap is related with the magnetic topological insulator rather than Weyl semimetal; in the caption of Fig.1, the authors state that Fig.1 shows “Zero magnetic field Chern states and transport properties.”, while Fig. 1a and 1b are calculation results of “spin-aligned $\text{Mn}(\text{Bi}_{1-x}\text{Sb}_x)_2\text{Te}_4$ ”. To avoid misleading, the authors should correct these statements.

3. In Line 214, the authors state that “We also note that since the position of Weyl points depends on the exchange splitting, ..., and thus may change the Chern numbers and the gaps in the absence of magnetic field”. In Line 248, the authors also state that “our transport results for different Sb concentrations highlight the importance of B=0 Weyl point separation for the Chern state’s quantization”. I’m confused by this statement since $\text{Mn}(\text{Bi}_{1-x}\text{Sb}_x)_2\text{Te}_4$ is an antiferromagnetic topological insulator under zero magnetic field and becomes ferromagnetic Weyl semimetal when applying moderate perpendicular magnetic field (above 7 T in this paper). Why the Weyl physics under strong magnetic field will influence the Chern number and gaps under zero magnetic field? Do the authors have any evidences for the existence of Weyl points under zero magnetic field?

4. Applying both top and back gate will generate an electric field in the sample and it has been shown that this electric field will influence the transport properties of MnBi_2Te_4 devices (Nature 595, 521–525 (2021)). Does the electric field have any influence on the sample properties in this paper, for example, changing the band structures?

5. The authors show the tunable Chern insulator states generated by the anomalous Landau Levels mainly in $\text{Mn}(\text{Bi}_{0.74}\text{Sb}_{0.26})_2\text{Te}_4$. Can the tunable Chern insulator states generated by the anomalous Landau Levels be observed in MnBi_2Te_4 devices?

6. The author state in Line 134 that “our 11-SL MnBi_2Te_4 (Fig. S5b) and 16-SL $\text{Mn}(\text{Bi}_{0.8}\text{Sb}_{0.2})_2\text{Te}_4$ (Fig. S7) reveal the developing of ν_x plateaus with higher Chern numbers near the CNPs”. The related experimental evidence, especially in Fig.S5, seems quite elusive.

7. The data quality of the main 18-SL $\text{Mn}(\text{Bi}_{0.74}\text{Sb}_{0.26})_2\text{Te}_4$ device seems not so good. For example, the C=1 and C=2 states shown in Extended Data Figure E2 don’t exhibit well-defined plateau and the Hall resistance is far away from the quantized value. What is the reason for this?

8. The authors state that the C=2 state in the 18-SL $\text{Mn}(\text{Bi}_{0.74}\text{Sb}_{0.26})_2\text{Te}_4$ device appears in CAFM phase. How the authors define the spin flip and spin flop magnetic field?

9. The authors claim that the anomalous Landau Levels originate from the interplay of Chern insulator states and Landau quantization. How to understand the interplay? Detailed discussions are absent in the current manuscript and more related analysis are desired for better understanding.

10. The color map in Fig. S4 is not intuitive. The resistance (R)-temperature (T) curves of $\text{Mn}(\text{Bi}_{1-x}\text{Sbx})_2\text{Te}_4$ would be helpful for understanding the surface-like and bulk-like behaviors.

11. Does the $C=1$ state also originate from the anomalous Landau Levels?

12. What do the blue lines in Fig.4 and red arrows in Extended Data Figure E3 represent?

13. Typos should be avoided. For example, “we report a tunable layer-dependent of $C=1$ state” in the abstract part should be “we report a tunable layer-dependent $C=1$ state” ; “the dualgate maps in Figs. 4e and f show a series of...” should be “the dualgate maps in Figs. 3e and f show a series of...” .

Responses to Reviewers' comments:

Reviewer #1 (Remarks to the Author):

This work discusses the interplay between the intrinsic Chern insulator state in $\text{MnBi}_{2-x}\text{Sb}_x\text{Te}_4$ with 3 different compositions and Landau quantization. Their conclusions seem 2-fold, first the thickness range over which the Chern insulator state is observed is extended to large thicknesses for high Sb content due to the changing Weyl node spacing. Second, an interplay between the Chern insulator state and Landau quantization gives rise to a set of anomalous Landau levels with an unconventional sequence of quantization index (Chern number). This work has the potential to be impactful and noteworthy but, in some instances, the main message of the manuscript is not clearly presented and diluted in detail. Because of this, I cannot yet tell if the second conclusion pertaining to the interpretation of the Landau quantized regime is well supported by data. Can the authors address the following issues:

Response: We thank the reviewer for providing his professional insight to help us improve the shortcoming in this manuscript.

- To better address the issue that I'm facing, can the authors construct a Landau fan diagram similar to the ones shown in Fig. 4 for each sample shown in Fig. 3?

Response: We follow the suggestions from the reviewer and have added the Landau fan diagram for the $\text{Mn}(\text{Bi}_{0.74}\text{Sb}_{0.26})_2\text{Te}_4$ samples at 21 and 14SL in the S.I. Fig. S4 & S5, respectively. We note that the Landau fan diagrams were constructed by assuming the spin-aligned phase over the entire magnetic field range. The reason for doing so is to determine the actual (zero-field) Chern number of the $\text{Mn}(\text{Bi}_{0.74}\text{Sb}_{0.26})_2\text{Te}_4$ at a given thickness (also refer to the response to Q1, Reviewer#2). Thus, it does not provide one-to-one matching to the experiment obtained Landau fan structure, especially at low magnetic fields. We also note that as the Chern states in $\text{Mn}(\text{Bi}_{0.74}\text{Sb}_{0.26})_2\text{Te}_4$ are strongly affected by the exchange coupling term, J_s , therefore, we included the Landau fan diagram for different J_s for comparison in these figures. In the following, we provide a more detailed interpretation of the quantization states for each $\text{Mn}(\text{Bi}_{0.74}\text{Sb}_{0.26})_2\text{Te}_4$ sample.

Noted that as we observed only the $C=1$ state in the 21SL $\text{Mn}(\text{Bi}_{0.74}\text{Sb}_{0.26})_2\text{Te}_4$ device, we anticipate that the reasons could be (i) the poor sample quality limits the access to higher quantization states, and (ii) the narrower anomalous LLs' gap for higher filling factors in thicker $\text{Mn}(\text{Bi}_{0.74}\text{Sb}_{0.26})_2\text{Te}_4$, as inferred theoretically in S.I. Fig. S5a.

We mainly focus the discussion on the 18SL and 14SL $\text{Mn}(\text{Bi}_{0.74}\text{Sb}_{0.26})_2\text{Te}_4$. The observation of the $C=2$ state and the reasonably good developed $C=3$ state at a high magnetic field is consistent with our model (Fig. 4b and extended data Fig. E3), indicating that the 18SL $\text{Mn}(\text{Bi}_{0.74}\text{Sb}_{0.26})_2\text{Te}_4$ lies in the Chern number=2. In comparison, the 14SL $\text{Mn}(\text{Bi}_{0.74}\text{Sb}_{0.26})_2\text{Te}_4$ presents a quite different feature from the 18SL $\text{Mn}(\text{Bi}_{0.74}\text{Sb}_{0.26})_2\text{Te}_4$ as (i) the Fermi level in 14SL $\text{Mn}(\text{Bi}_{0.74}\text{Sb}_{0.26})_2\text{Te}_4$ sits between $C=0$ and $C=1$ state, while the 18SL sits around the $C=2$ state. (ii) the 14SL $\text{Mn}(\text{Bi}_{0.74}\text{Sb}_{0.26})_2\text{Te}_4$ exhibits a better developed $C=1$ state, while the 18SL $\text{Mn}(\text{Bi}_{0.74}\text{Sb}_{0.26})_2\text{Te}_4$ show better developed $C=3$ state. (iii) the LL fan diagram in 14SL MBST appears similar to the non-magnetic, $J_s=0$ case (refer to Fig. 4a $J_s=0$ for 18SL

$\text{Mn}(\text{Bi}_{0.74}\text{Sb}_{0.26})_2\text{Te}_4$), indicating that the higher LLs $C=3, 5$, etc. belongs to the non-anomalous LLs (black curves), whereas the $C=1$ is the Chern insulator state. Combining these observations, we infer that the 14SL $\text{Mn}(\text{Bi}_{0.74}\text{Sb}_{0.26})_2\text{Te}_4$ sits in Chern number=1, which is similar to the Chern number of $\text{Mn}(\text{Bi}_{0.8}\text{Sb}_{0.2})_2\text{Te}_4 < 12\text{SL}$ and $\text{MnBi}_2\text{Te}_4 < 9\text{SL}$.

However, our calculation shows that a reasonably large gap of $C=2$ anomalous LL should develop in the spin-alignment phase. To check this, we examine their quantization at a higher magnetic field up to 18T. Unfortunately, our 14SL $\text{Mn}(\text{Bi}_{0.74}\text{Sb}_{0.26})_2\text{Te}_4$ sample did not survive in the first cooling, therefore we added the measurement results for 11SL and 10SL $\text{Mn}(\text{Bi}_{0.8}\text{Sb}_{0.2})_2\text{Te}_4$ (S.I. Fig. S16) which appear to be in the Chern number=1 as the 14SL $\text{Mn}(\text{Bi}_{0.74}\text{Sb}_{0.26})_2\text{Te}_4$. Similar to the 14SL MBST, the $C=1$ and 3 states can also be resolved in the medium field ($\sim 10\text{T}$) for the 11 and 10SL $\text{Mn}(\text{Bi}_{0.8}\text{Sb}_{0.2})_2\text{Te}_4$. We show the development of the $C=2$ state at the higher magnetic field in both the 11 and 10SL $\text{Mn}(\text{Bi}_{0.8}\text{Sb}_{0.2})_2\text{Te}_4$ (S.I. Fig. S16). Note that this LL sequence is unusual as compared to a non-magnetic TI (refer to Fig. 4a), where the odd LL sequence should preserve without splitting even at the high magnetic field. The well-developed $C=2$ state over a relatively wide gate voltage range suggests the anomalous LLs in contrast to the $C=3$ and above states.

- The thickness of the flakes shown in Fig. 3 is somewhat large compared to what is typically studied in the literature, and raises the question of whether the two gates are actually able to deplete the bulk (especially in the case of the 21-SL sample). Is it possible that in this sample the authors are controlling the amount of band bending at the top and bottom surface?

Response: We agree with the reviewer that the dual gate voltages can, to some extent, control the band bending at the top and bottom surfaces. However, we believe that in the gate voltage range, we are still able to control the surface charge density. This is also supported by the field-dependent line profiles for the 21-SL $\text{Mn}(\text{Bi}_{0.74}\text{Sb}_{0.26})_2\text{Te}_4$ samples depicted in Fig. S11, where we can see the carrier type changes when applying gate voltages across the charge neutrality.

- Line 91: Is the gate dielectric made out of mica or hBN as described in the methods section. If it's mica please describe the process by which mica was transferred in the methods section.

Response: We thank the reviewer for pointing out the inconsistency. We have corrected the gate dielectric part and added the type of mica and the transfer process in the method section of the revised manuscript.

- The theory predicts a specific behavior as a function of positive versus negative magnetic field: "When the magnetic field is reversed, the spin of the anomalous LLs is reversed, but the sign of the exchange coupling is also reversed, so the sign of the filling factor range at which strong quantum Hall effects [sic] will not change." Is that consistent with the data? Can the authors show data where this is compared?

Response: We clarify that the statement was not written appropriately. We intend to say that the Chern number depends only on the direction of the magnetic moment, which is determined by the direction of the magnetic field. We have corrected the statement to “When the magnetic field is reversed, both the spin of the anomalous LLs and the sign of the exchange coupling between the local moments and the Dirac electrons are reversed, therefore the sign of the filling factor will follow the sign of the magnetic field” in the revised manuscript.

- The authors claim to observe a signature of the $C=1$ filling shifting below the Fermi level and that the surface gap at the Fermi level is at filling factor of higher Chern number, but how can they specifically determine the position of the surface gap in their measurement? For instance, in Fig. 3i, both $C=0$ and $C=1$ overlap with the minimum in conductivity, making the precise position of the surface gap (or the charge neutrality point) ambiguous versus gate voltage.

Response: We clarify that the assignment of the charge neutrality point (CNP) is based on the gate-dependent resistivity peak (or minimum conductivity) at zero magnetic field. We further show that this is in agreement with the analysis of the ordinary Hall effect at the low magnetic field (S.I. Fig. S11-S13) where the positive and negative slopes corresponding to the hole and electron carriers, respectively, and the charge neutrality region sits in between. For the 14-SL $\text{Mn}(\text{Bi}_{0.74}\text{Sb}_{0.26})_2\text{Te}_4$ sample, we also show that the development of the Landau levels as shown in R_{xx} and R_{yx} (S.I. Fig. S15) can be traced linearly in magnetic field down to zero magnetic field, corresponding to its charge neutrality point. However, we agree with the reviewer that assigning a precise position of gate voltage to the surface gap or CNP can be ambiguous as we are dueling with a disordered system. We emphasize that this analysis is not meant to be a precise CNP position, but the CNP's relative gate position to the Chern states.

- Line 231 to 237: the authors compare their findings in the 18-SL sample from Fig. 3 with theory, referring to Fig. S3, but Fig. S3 does not include a calculation for 18-SLs.

Response: We apologize for citing the wrong figure. The cited Fig. S3 should be replaced by Extended Data Fig. E3. We corrected the mistake in the revised manuscript.

- I am having trouble deciphering the color maps in the Fig. 2. For instance in Fig. 2(c) does the dark red +/- 6T correspond to a region with very high Hall resistivity or zero Hall resistivity? Or is that showing mixing with the magnetoresistance?

Response: We clarify that the high Hall resistivity in the color maps in Fig. 2a-c is a result of mixing with the magnetoresistance. The antisymmetrized R_{yx} (Fig. 2d-f) in the line cuts show the minimum R_{yx} in the region. We added a description in the caption of Figure 2 to clarify it in the revised manuscript.

- The Hall resistivity plateau in Fig. 2f is very wide, and seems robust up to more than 4T. This behavior seems inconsistent with the magnetic phase boundaries shown in Fig. 2c. Is this behavior understood?

Response: We clarify that the Hall resistivity plateau in Fig. 2f was taken at a particular gate voltage where the R_{yx} is maximized (quantized to $C=1$ state) in the FM phase. The full dataset of the gate-dependent Hall resistivity plateau is added in Fig. S11 in the revised manuscript. We clarify that the magnetic phase boundaries can still be seen in the R_{xx} curve where the “jump” at $\pm 2.5T$ is corresponding to the spin-flop transition. However, the origin of the wide and robust high resistive plateau in the hole conduction region is another interesting subject for future studies as it could be related to the axion insulator state.

- There needs to be a more intuitive explanation of the meaning of “anomalous Landau levels” and how they determine the value of the Chern number. Is it due to a changing number of inverted bands driven by spin-splitting or is it something more complex?

Response: We thank the suggestion from the reviewer. We have included a more detailed description of the derivation of the anomalous Landau levels and the determination of their Chern number. We clarify that the basic theory of the anomalous Landau levels was equivalent to a generalized Su-Schrieffer-Heeger (SSH) model as represented by equation (3) in the S.I. The typical wavefunction distribution of the $n=0$ anomalous LLs (refer to S.I. Fig. S2 for the details) bands localized at the surface and into the bulk are corresponding to the red and blue lines, respectively, in the LL spectra shown in Fig. 4a, c, and d. The anomalous LLs were distinguished from the ordinary LL band (non-anomalous) by their non-dispersive band in magnetic field, similar to $n=0$ LLs in graphene and 3D TIs in non-magnetic cases. The band energies of anomalous LLs with respect to the Fermi level depend on the exchange coupling parameter J_s and the spin directions. This causes an imbalance between the total number of subbands (including anomalous and non-anomalous LL bands) above and below the Fermi level and eventually gives rise to the non-zero Chern number at the Fermi level (QAH phase at zero field or Chern insulator state at an applied field). Subsequently, the Chern number for the $n=0$ LLs with higher band index (as indexed in Fig. 4a, c, and d) can be determined by $(N_{E<E_F} - N_{E>E_F})/2$ where $N_{E<E_F}$ ($N_{E>E_F}$) is the total number of subbands below (above) Fermi level.

- Why are $J_s=30\text{meV}/34\text{meV}$ chosen to compare with data?

Response: We clarify that the value of $J_s=34\text{meV}$ is determined by DFT calculations for perfect $\text{Mn}(\text{Bi}_{0.74}\text{Sb}_{0.26})_2\text{Te}_4$ thin film. However, the exchange coupling, J_s , can be weakened due to defects and surface degradation. We found that our data match better for smaller J_s , which also indicates the imperfection in the $\text{Mn}(\text{Bi}_{0.74}\text{Sb}_{0.26})_2\text{Te}_4$ thin films due to the crystal growth, device fabrication, and/or degradation in the samples. In Fig. 4c and d, the two different J_s values of 34 and 30 meV were chosen to demonstrate that the anomalous Landau levels developed at the high magnetic spin-alignment phase are more robust to the change in J_s than the zero magnetic

field spin-alignment phase.

Reviewer #2 (Remarks to the Author):

MnBi₂Te₄ and its family of materials have brought great progress in studying novel topological quantum states. Chong et al. study the Chern insulator states in the intrinsic magnetic topological material Mn(Bi_{1-x}Sb_x)₂Te₄ devices and their relationship with the device thickness and Sb concentrations. Firstly, they find that the thickness range for the C= 1 state shows a correlation with the Sb concentrations and they attribute this behavior to the change of separation of Weyl points induced by Sb substitution. Then, their transport results for different Sb concentrations suggest the existence of anomalous Landau levels in Mn(Bi_{1-x}Sb_x)₂Te₄, which are attributed to the interplay of B=0 Chern insulator state and Landau quantization. These interesting results indicate that Mn(Bi_{1-x}Sb_x)₂Te₄ devices may provide a tunable platform for probing the physics of new topological states, which may stimulate further theoretical and experimental exploration of nontrivial topology in MnBi₂Te₄ and its family of materials. Overall I recommend publication, provided that the authors address the following questions:

Response: We thank the reviewer for appreciating this work and for the important comments and suggestions to help us improve this manuscript.

1. The authors attribute the anomalous Landau Levels to the interplay of B=0 Chern insulator state and Landau quantization. However, no Chern insulator state can be experimentally observed under B=0 in this paper. How to understand the so-called B=0 Chern insulator state theoretically?

Response: We clarify that the B=0 Chern insulator state is determined under the assumption of the spin-alignment phase at zero magnetic field. The reason for the assumption is to determine the actual Chern number of the magnetic exchange gap in the Mn(Bi_{0.74}Sb_{0.26})₂Te₄. As illustrated in the high magnetic field spin-alignment phase, the Chern insulator state becomes elusive when more Chern number states develop in the presence of anomalous and normal Landau levels. To avoid confusion, we have added a note “Noted that for the convenience (simplicity) in calculations, we ignore the magnetic transitions and assume the spin-aligned phase for all magnetic field ranges. This assumption enables the determination of the actual Chern number due to the band topology in the limit of zero magnetic field and the development of anomalous and non-anomalous LLs under high magnetic field.” to clarify it.

2. Some key statements in this paper are improper, unclear and puzzling. For example, in Line44, the authors state that “We show that Sb substitution extends the surface gap regime to a wider thickness range by suppressing conduction from the trivial bulk bands, enabling better access to their Weyl physics”, which is puzzling since the surface gap is related with the magnetic topological insulator rather than Weyl semimetal; in the caption of Fig.1, the authors state that Fig.1 shows “Zero magnetic field Chern states and transport properties.”, while Fig.1a and 1b are calculation results of “spin-aligned Mn(Bi_{1-x}Sb_x)₂Te₄”. To avoid misleading, the authors should correct these statements.

Response: We thank the reviewer for pointing out the misleading statements. We have followed the suggestion from the reviewer and revised the following statements: “We show that Sb substitution extends the surface gap regime to a wider thickness range by suppressing conduction from the trivial bulk bands” and changed the Fig. 1 caption to “Tunable Chern insulator states and transport properties” in the revised manuscript.

3. In Line 214, the authors state that “We also note that since the position of Weyl points depends on the exchange splitting, ..., and thus may change the Chern numbers and the gaps in the absence of magnetic field”. In Line 248, the authors also state that “our transport results for different Sb concentrations highlight the importance of B=0 Weyl point separation for the Chern state’s quantization”. I’m confused by this statement since $\text{Mn}(\text{Bi}_{1-x}\text{Sb}_x)_2\text{Te}_4$ is an antiferromagnetic topological insulator under zero magnetic field and becomes ferromagnetic Weyl semimetal when applying moderate perpendicular magnetic field (above 7T in this paper). Why the Weyl physics under strong magnetic field will influence the Chern number and gaps under zero magnetic field? Do the authors have any evidences for the existence of Weyl points under zero magnetic field?

Response: We again apologize for the unclear statement. We clarify again that the focus of this manuscript is on the magnetic field-induced spin-alignment phase. The zero magnetic field description is mainly for the theoretical perspective to distinguish the actual Chern number of the system from the anomalous Landau levels developed at high magnetic field. We revised the statement to “Our transport results for different Sb concentrations highlight the importance of the Weyl point separation in the spin-aligned magnetic phase to the Hall quantization.” in the conclusion part of the revised manuscript.

4. Applying both top and back gate will generate an electric field in the sample and it has been shown that this electric field will influence the transport properties of MnBi_2Te_4 devices (Nature 595, 521–525 (2021)). Does the electric field have any influence on the sample properties in this paper, for example, changing the band structures?

Response: As our dualgated samples were made of the thicker MBST flakes, the electric field effect (as shown by the bending in the σ_{xx} plateau minima at higher electric field region) is not as clear as the thin MnBi_2Te_4 samples like the reference (Nature 595, 521-525 (2021)) which the reviewer cited in the comment. However, this will be an interesting topic to follow up on the thinner MBST devices in our future studies.

5. The authors show the tunable Chern insulator states generated by the anomalous Landau Levels mainly in $\text{Mn}(\text{Bi}_{0.74}\text{Sb}_{0.26})_2\text{Te}_4$. Can the tunable Chern insulator states generated by the anomalous Landau Levels be observed in MnBi_2Te_4 devices?

Response: The anomalous Landau levels can also be observed in MnBi_2Te_4 under similar

conditions as (i) a higher magnetic field to push away the ordinary Landau levels away from the Fermi level. (ii) MnBi_2Te_4 thin film with larger thickness to host a larger number of anomalous Landau level bands. Indeed, we have also observed the signature of anomalous Landau levels in our MnBi_2Te_4 samples, which will be reported in a follow-up work.

6. The author state in Line 134 that “our 11-SL MnBi_2Te_4 (Fig. S5b) and 16-SL $\text{Mn}(\text{Bi}_{0.8}\text{Sb}_{0.2})_2\text{Te}_4$ (Fig. S7) reveal the developing of ρ_{yx} plateaus with higher Chern numbers near the CNPs”. The related experimental evidence, especially in Fig.S5, seems quite elusive.

Response: We thank the reviewer for pointing out the elusive statement. We have decided to remove the sentence from our revised manuscript as it does not carry conclusive information.

7. The data quality of the main 18-SL $\text{Mn}(\text{Bi}_{0.74}\text{Sb}_{0.26})_2\text{Te}_4$ device seems not so good. For example, the $C=1$ and $C=2$ states shown in Extended Data Figure E2 don't exhibit well-defined plateau and the Hall resistance is far away from the quantized value. What is the reason for this?

Response: We agree with the reviewer that the 18-SL $\text{Mn}(\text{Bi}_{0.74}\text{Sb}_{0.26})_2\text{Te}_4$ is not fully quantized in certain Chern numbers, for example, the $C=1$ and $C=2$ states, suggesting the exchange gap is not fully developed at the base temperature of 2K. To further verify the Chern states, we constructed the flow diagram plots in σ_{xx} and σ_{xy} space at the two studied magnetic fields, 6T and 9T, as included in S.I. Fig. S14 in the revised manuscript. As shown in the flow diagram plots, the $C=1$ and 2, and $C=1$ and 3 states can be seen at magnetic fields of 6T and 9T, respectively. As the $C=3$ state quantized reasonably well in this sample, the bulk conduction channels might not be the dominant issue for the poor quantization in other states. We anticipate that the smaller gap sizes of these poor quantization state relative to the disorder are the main reason.

8. The authors state that the $C=2$ state in the 18-SL $\text{Mn}(\text{Bi}_{0.74}\text{Sb}_{0.26})_2\text{Te}_4$ device appears in CAFM phase. How the authors define the spin flop and spin flip magnetic field?

Response: We clarify that the spin flop and spin flip magnetic field were defined as the magnetic field for the transitions from antiferromagnetic to canted antiferromagnetic phases, and canted antiferromagnetic to ferromagnetic phases. We added the definition at their first appearance in the main text of the revised manuscript. We also clarify that the spin flop and spin flip magnetic fields for the 18-SL $\text{Mn}(\text{Bi}_{0.74}\text{Sb}_{0.26})_2\text{Te}_4$ are determined from the kinks in the magnetic field-dependent R_{xx} and R_{yx} curves as indicated by the blue and red arrows, respectively, in the S.I. Fig. S12. The magnetic transition fields are consistent with the observation in their bulk parent compounds.

9. The authors claim that the anomalous Landau Levels originate from the interplay of Chern insulator states and Landau quantization. How to understand the interplay? Detailed discussions are absent in the current manuscript and more related analysis are desired for better understanding.

Response: We thank the reviewer for pointing out the missing discussions on the interplay of the Chern insulator states and the Landau levels. We clarify that the Chern insulator state is referring to the actual Chern number of the system, which we define theoretically as the state in the spin-aligned phase without the application of magnetic field. In the presence of magnetic field, a series of field-dependent Landau levels and field-independent anomalous Landau levels will develop. Our calculations show that the Chern insulator state and anomalous Landau levels can stabilize at the Fermi levels as the Landau level bands expand in magnetic field. Although the anomalous Landau levels originated from the $n=0$ LLs are field independent, the energy spacings or the gap sizes of the Chern insulator states and anomalous Landau levels depend on the spacing between themselves and the ordinary Landau levels. Thus, the Landau quantization in MBT based system is complicated by the existing and the interplay of these Landau level bands. We have added a brief discussion in the revised manuscript.

10. The color map in Fig. S4 is not intuitive. The resistance (R)-temperature (T) curves of $\text{Mn}(\text{Bi}_{1-x}\text{Sbx})_2\text{Te}_4$ would be helpful for understanding the surface-like and bulk-like behaviors.

Response: We follow the suggestion from the reviewer and have added the resistance-temperature curves for the MnBi_2Te_4 and $\text{Mn}(\text{Bi}_{0.74}\text{Sb}_{0.26})_2\text{Te}_4$ in the S.I. Fig. S7 in the revised manuscript displays the insulating surface-like and metallic bulk-like behaviors.

11. Does the $C=1$ state also originate from the anomalous Landau Levels?

Response: We clarify that the $C=1$ state has been widely referred to as the Chern insulator state because most of the thin film MnBi_2Te_4 reported in the literature (3-8SL) lies in the $C=1$ Chern number. Our calculations show that even in thicker MnBi_2Te_4 or MBST film with a higher Chern number, the $C=1$ state can still exist in the form of anomalous Landau levels and we can track down their origin from our theoretical models.

12. What do the blue lines in Fig.4 and red arrows in Extended Data Figure E3 represent?

Response: The blue lines in Fig. 4 represent the $n=0$ anomalous Landau levels, whereas the red arrows in Extended Data Fig. E3 denote Chern number determined theoretically from the zero magnetic spin alignment phase. We added the descriptions in the revised manuscript.

13. Typos should be avoided. For example, “we report a tunable layer-dependent of $C=1$ state”

in the abstract part should be “we report a tunable layer-dependent $C=1$ state” ; “the dualgate maps in Figs. 4e and f show a series of...” should be “the dualgate maps in Figs. 3e and f show a series of...”.

Response: We appreciate the reviewer for pointing out the typos. We have corrected them in the revised manuscript.

REVIEWERS' COMMENTS

Reviewer #1 (Remarks to the Author):

The authors have answered my questions. I can now recommend this manuscript for publication.

Reviewer #2 (Remarks to the Author):

All my concerns have been properly addressed. I recommend the publication.